---

**RESEARCH ARTICLE**                                     **TECHNIQUES AND RESOURCES**

# Automated, high-throughput *in situ* hybridization of sea urchin (*Lytechinus pictus*) embryos

**Yoon Lee, Chloe Jenniches, Rachel Metry, Gloria Renaudin, Svenja Kling, Evan Tjeerdema, Elliot W. Jackson and Amro Hamdoun***

## ABSTRACT

Despite the reach of *in situ* hybridization (ISH) in developmental biology, it is rarely used at scale. The major bottleneck is the throughput of the assay, which relies upon labor-intensive manual steps. The goal of this study was to develop a high-throughput, automated hybridization chain reaction (HCR) pipeline for the sea urchin (*Lytechinus pictus*). Our method, which we term high-throughput (HT)-HCR, can process 192 gene probe sets on whole-mount embryos within 32 h. The physical properties of sea urchin embryos enabled us to utilize a 96-well plate format, miniaturized reaction volumes, a general-purpose robotic liquid handler and automated confocal microscopy. Using this approach, we produced high quality localization data for 101 target genes across three developmental stages. The results reveal the localization of previously undescribed physiological genes, as well as canonical developmental transcription factors. HT-HCR represents an order of magnitude increase in the throughput of spatial expression profiling studies utilizing the sea urchin. This will enable more-sophisticated perturbation analyses and drug-screening efforts in this emerging animal model.

KEY WORDS: *Lytechinus pictus,* Hybridization chain reaction (HCR), High-throughput, *In situ* hybridization (ISH), Transcription factor, Transporter, Sea urchin, Spatial transcriptomics

## INTRODUCTION

*In situ* hybridization (ISH) is a foundational method in developmental biology. In sea urchins, it has been the basis for understanding cell fate, identity and function (Juliano et al., 2006, 2010; Luo and Su, 2012), as well as for the description of some of the most detailed gene regulatory networks of development (Angerer and Angerer, 1981; Davidson et al., 2002). These studies take advantage of the tight development synchrony of this animal model, which makes it well suited for the study of spatial and temporal control of gene expression during development.

Yet despite this long history in the field (Angerer and Angerer, 1981; Cheatle Jarvela et al., 2016; Croce et al., 2006; Davidson et al., 2002; Golconda et al., 2019; Juliano et al., 2010; Katow et al., 2004;

Center for Marine Biotechnology and Biomedicine, Scripps Institution of Oceanography, University of California San Diego, La Jolla, CA 92037, USA.

*Author for correspondence (ahamdoun@ucsd.edu)

Y.L., 0000-0003-0147-3208; C.J., 0009-0000-9955-393X; R.M., 0000-0002-6156-5799; G.R., 0009-0009-7909-7733; S.K., 0000-0001-5109-2521; E.T., 0000-0003-3732-9232; E.W.J., 0000-0002-4213-3851; A.H., 0000-0003-2568-048X

Lee et al., 2023; Perillo et al., 2016; Rodríguez-Sastre et al., 2023; Schrankel and Hamdoun, 2021; Sharma and Ettensohn, 2011; Wood et al., 2018), ISH in sea urchins remains a tedious manual procedure, often requiring years of effort to characterize large gene sets (Valencia and Peter, 2024). This has made the application of the technique prohibitive for large-scale screens of gene expression patterns, e.g. as required to generate developmental gene expression atlases or for perturbation analyses.

Hybridization chain reaction (HCR) is based on the use of affordable synthetic short (45 nucleotides) DNA oligonucleotide probes that innately suppress background and enable single-step multiplexing (Choi et al., 2014, 2018). These qualities facilitate automation and increased throughput in two ways. First, HCR probes are easily commercially synthesized, within a week, as pools of unmodified oligonucleotides containing a set of short probes specific to several (typically three to five) gene targets, using a synthesis process comparable to primer synthesis. This allows researchers to quickly and affordably obtain probes for numerous targets, a vastly more-efficient approach than production of long RNA probes. Second, automation of both multiplexing and hybridization incubation periods is made possible by the mechanism of signal amplification. Specifically, multiplexed signal amplification is enabled by using sequence-specific reagents to drive discrete hybridization chain reactions of different channels of fluorescent oligonucleotides specific to their respective gene targets. This eliminates the need for many of the sequential multiplexing steps that are common to other ISH methods. In addition, the amplification of the signal is driven by the sequence complementarity of fluorescent oligonucleotides in a controlled molecular crowdant buffer, reducing the possibility of nonspecific signal and overstaining.

The goal of this study was to develop a robust pipeline for high-throughput sea urchin ISH that can be easily applied by end users. Previous efforts to automate ISH have typically relied upon bespoke hardware solutions, with limited throughput and utility in other assays. Our approach relies on plate and robotic formats likely to be found in many core facilities or individual labs. These general liquid handlers require smaller capital expenditure risk, since the instrument can also perform a variety of other routine tasks. Furthermore, the capability to complete the assay in standard plate formats allow the automation of image data acquisition using automated confocal imaging systems commonly used in high-content drug-screening assays (Boutros et al., 2015; Dranchak et al., 2023; Stossi et al., 2023).

Here, we present high-throughput HCR (HT-HCR), a fully automated miniaturized HCR pipeline capable of running 192 probe sets in 32 h. Using HT-HCR, we probed three early developmental stages of sea urchin (*Lytechinus pictus*) embryos, with a probe panel designed to target genes expressed at different developmental stages and in different spatial domains, and representing a wide range of functional categories. The results resolved the localization patterns of 101 sea urchin genes, belonging to multiple KEGG categories

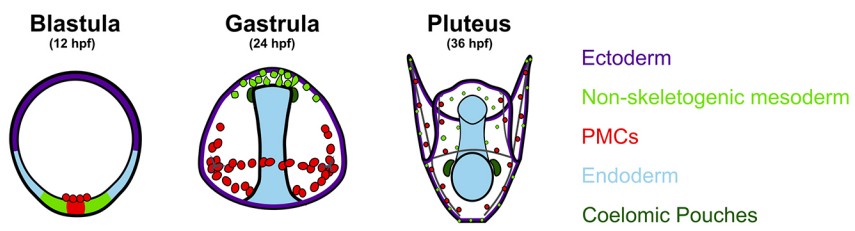

**Fig. 1. Diagram of a fully automated HCR workflow.** (A) Automated sample processing using the Opentrons Flex robot. (A1) The consumable probe plate as the starting vessel. A 96-well plate, where each well contains 5 µl of a pool of probes for four mRNA targets is used as the starting vessel for sample processing. For all liquid handling steps throughout the protocol, dispense speeds are set at 200 µl/s and aspiration speeds at 10 µl/s. All aspirations are set to use a tip to well bottom distance of 3 mm. (A2) The probe hybridization step. 10 µl of fixed embryos in hybridization buffer are transferred to each well to start the probe hybridization step. The inset illustrates transferability of sea urchin embryos using standard automation pipette tips. (A3) The probe wash step. Probes are washed from the sample using a formamide-based probe wash buffer. The insets illustrate the liquid handling speeds and tip to bottom distances allow for resuspension and mixing of embryos throughout washes. (A4) The amplification step. 10 µl of embryos are transferred to a plate of pre-plated amplifier hairpins in dextran sulfate. (A5) The amplifier wash step. Amplifier hairpins are washed using 5×SSCT, and 150 µl of samples in 5×SSCT are subsequently transferred to a glass bottom high-content imaging plate. Samples are centered using a plate shaker module set at 450 rpm for 30 min. (B) Automated confocal image acquisition using the ImageXpress HT.ai. The imaging plates containing centered samples are placed into the confocal microscope. Depending on data storage availability, the center 4 or 9 are imaged for each well. An example of an output of the demultiplexed and multiplexed versions of a single site of a positive control sample well is shown. (C) A schematic of developmental stages imaged in this study to be used as a developmental staging key. PMCs, primary mesenchyme cells.

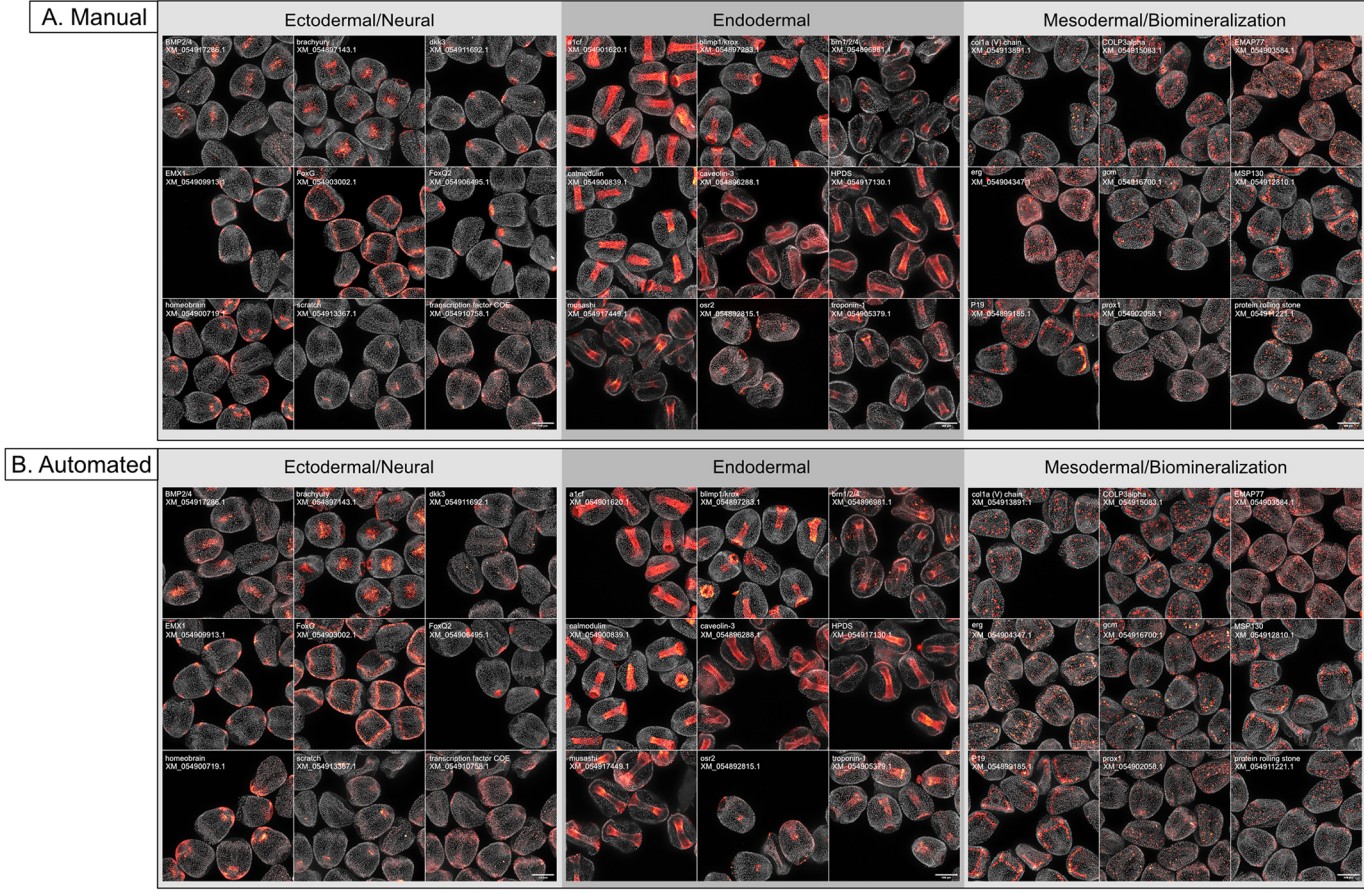

**Fig. 2. Comparison of results from manual and automated assays.** (A,B) Images show the localization of several positive control targets in 24 hpf embryos. Manually processed samples (A) and robotically processed samples (B) appear identical. Nuclear staining using Hoechst 33342 appears in gray. Scale bars: 100 μm.

(Kanehisa, 2019; Kanehisa and Goto, 2000; Kanehisa et al., 2023), including cell motility (2), cytoskeleton (4), genetic information processing (5), matrix proteins (3), membrane trafficking (3), membrane transport (18), metabolism (5), signal transduction (12), transcription factors (31) and transport/catabolism (2), and 16 categorized as 'other'.

## RESULTS

This HT-HCR screen approach produced localization data for 101 genes in *Lytechinus pictus*. The images are publicly available on our laboratory data repository, which is available through the following link Hamdoun Lab Image Repository. These included genes with a broad range of biological functions and localization patterns along the KEGG BRITE in *L. pictus* (Fig. S1, Table S1).

### Validation of the automated assay

As with manual approaches, not all genes targeted by HCR in our assay produced clearly interpretable images. In this case, from 192 targets, 101 provided clear expression patterns, while the other 91 lacked clear localizations (Table S2), potentially indicating weak or diffuse expression of the target gene. The images from these 91 ranged from appearing identical to negative controls to faint speckled signal. For example, there is a complete absence of signal for *MOB kinase 1A* (Fig. S2A,B), consistent with negative controls, and a faint speckled signal for *titin* (Fig. S2C), which may indicate weak ectodermal expression or simply be a negative. To determine whether these negative results might relate to probe concentration,

we performed the assay for three genes from our list of positive results [*tubulin alpha chain-like*, *COLP3alpha* and *collagen 1 alpha (V) chain*], and three from the negative results (*apmap*, *MOB kinase activator 1A* and *titin*) across four different concentrations (50 nM, 5 nM, 0.5 nM and 0.05 nM) (Fig. S3). All genes tested from our list of negative results produced negative or unclear localizations across all concentrations, and appeared similar to negative control samples (Fig. S3).

To test the robustness of our automated HCR protocol (Fig. 1), we optimized our liquid handling steps for accurate miniaturized sample volumes and maintenance of sample morphological integrity. Our optimization also considered efficient consumable usage (i.e. tip reuse steps) without the risk of sample contamination events. This was important for a fully automated protocol on a basic liquid handler because deck space limitations typically require users to be present to refill consumables during a paused state during the protocol run. We compared the results of the finalized protocol to those produced by manual HCR across stages and genes. One of our concerns was the maintenance of morphological integrity using automated liquid handling. We found that unmodified automation tips were able to transfer embryos in dense concentrations in a viscous hybridization buffer with little to no damage to 24 hpf embryos, a stage that is commonly used to study gastrulation (Fig. 2). We also tested the automated HCR protocol across stages, and demonstrated it is effective (Fig. S4) using probes for the genes *white protein* (Fig. S4A), *collagen 1-alpha(V) chain* (Fig. S4B) and *tubulin alpha 1* (*tuba1*) (Fig. S4C).

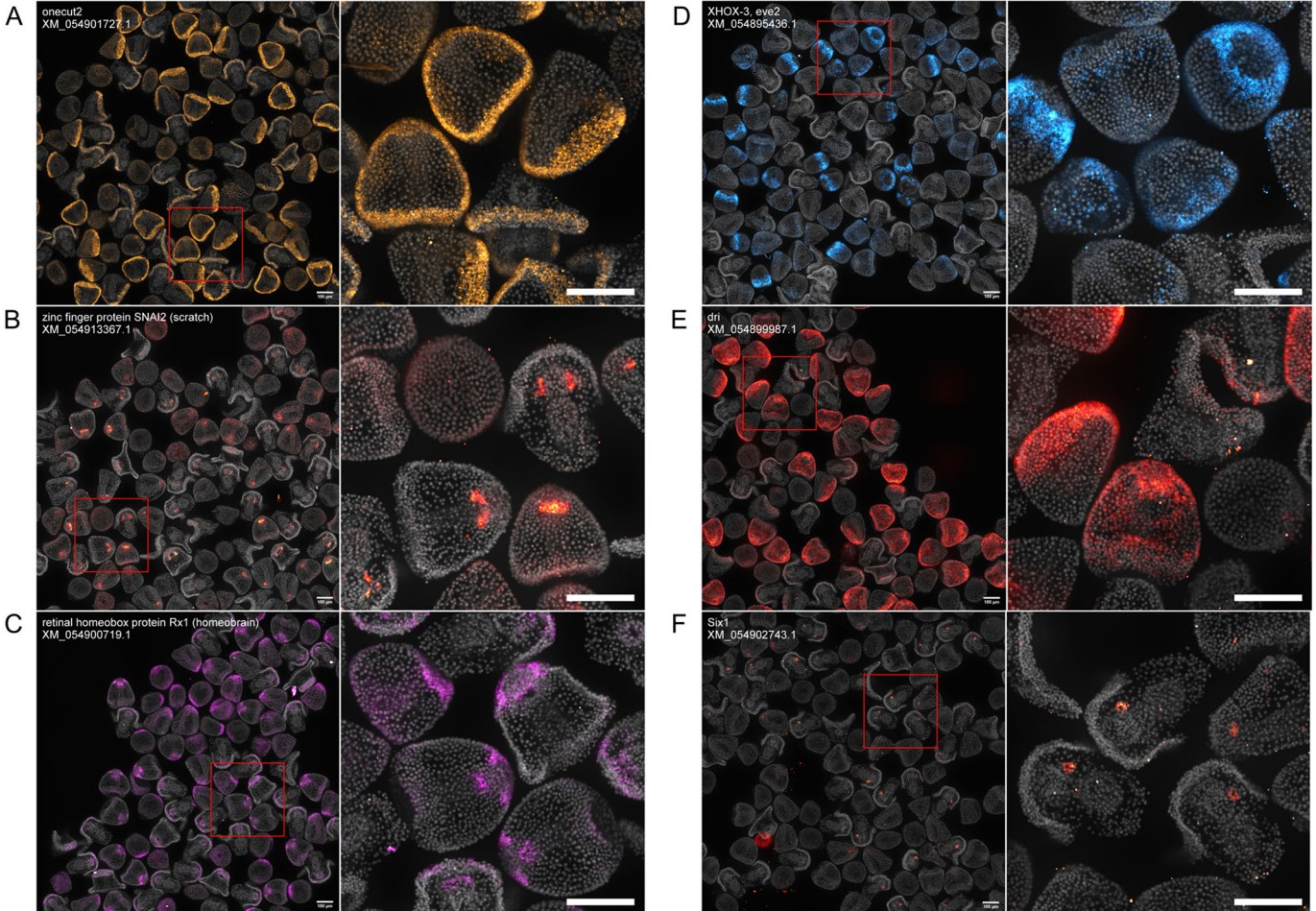

**Fig. 3. Validation of HT-HCR by localization of previously described transcription factor genes.** All images contain a mix of 12, 24 and 36 hpf embryos, and are automatically demultiplexed and stitched outputs resulting from an automated image processing workflow. Each image contains between 50 and 100 embryos, which are annotated with the NCBI RefSeq ID for the mRNA sequence targeted by probe oligonucleotides. (A) *onecut2*: expression is sparse in 12 hpf embryos. In 24 hpf and 36 hpf embryos, expression is concentrated in the ciliary band. (B) *scratch*: expression is either sparse or absent in 12 hpf embryos. At 24 hpf, expression is concentrated in the archenteron. At 36 hpf, expression is restricted to the coelomic pouches. (C) *homeobrain*: at 12 hpf, expression is present in the apical ectoderm. At 24 and 36 hpf, expression is in the apical ectoderm and stomodeum. In more developed 36 hpf embryos, expression can be seen in the upper region of the developing oral hood. (D) *XHOX-3* and *eve2*: expression is most enriched in 12 hpf embryos in a ring of cells in the aboral ectoderm. Sparse expression is visible in some 24 hpf embryos. (E) *dri*: expression is most enriched in 24 hpf embryos and is restricted to the oral ectoderm. Sparse expression is visible along the ciliary band in 36 hpf embryos. (F) *six1*: expression is restricted to the coelomic pouches in 36 hpf embryos. Scale bars: 100 µm.

To determine whether our automated assay would be able to produce similar results to manual approaches across spatial domains, we compared outputs of our assay to those for 27 genes (Fig. 2). We found that, qualitatively, the automated assay produced results comparable to those obtained by a manual approach. A comparison of the localization of three genes *empty spiracles homeobox 1* (*EMX1*), *hematopoietic prostaglandin D-synthase* (*HPDS*) and *P19* (previously shown by Costa et al., 2012) across three different automated runs indicated that these results were produced consistently across batches and runs (Fig. S5).

Next, we examined the efficacy of our automated protocol for mixed stage samples, focusing on transcription factors that were well described in urchins and expressed in highly spatially restricted patterns (Fig. 3). Overall, we found that unique morphological features of these three developmental stages allowed the easy differentiation of embryos across stages. Blastula stage embryos (12 hpf) were hollow spheres of cells, gastrula stage embryos (24 hpf) contained gut tubes and prism stage embryos (36 hpf)

possessed two distinct arms. Given this, we proceeded to perform the automated assay using mixed stage samples to conserve reagents, consumables and the number of runs.

Three of the localized transcription factors included *onecut2*, *scratch* and *homeobrain* (Fig. 3A-C), which are associated with neurodevelopment. The expression of *onecut2* is concentrated in the ciliary band in 24 and 36 hpf embryos, but is sparse in 12 hpf embryos (Fig. 3A). These patterns are similar to the localization patterns found in *Strongylocentrotus purpuratus* (Poustka et al., 2004). The expression of *scratch* in *L. pictus* appears different from the localization pattern found in *Lytechinus variegatus* (Slota et al., 2019), which showed expression that was concentrated in a few apical cells at 24 hpf. In *L. pictus*, the expression of *scratch* appears to be absent in 12 hpf embryos, and present in the foregut at 24 hpf and the coelomic pouches at 36 hpf (Fig. 3B). This could be due to some species-specific difference in localization or developmental timing. The expression of *homeobrain* recapitulates the localization patterns found in another urchin species, *Hemicentrotus pulcherrimus*,

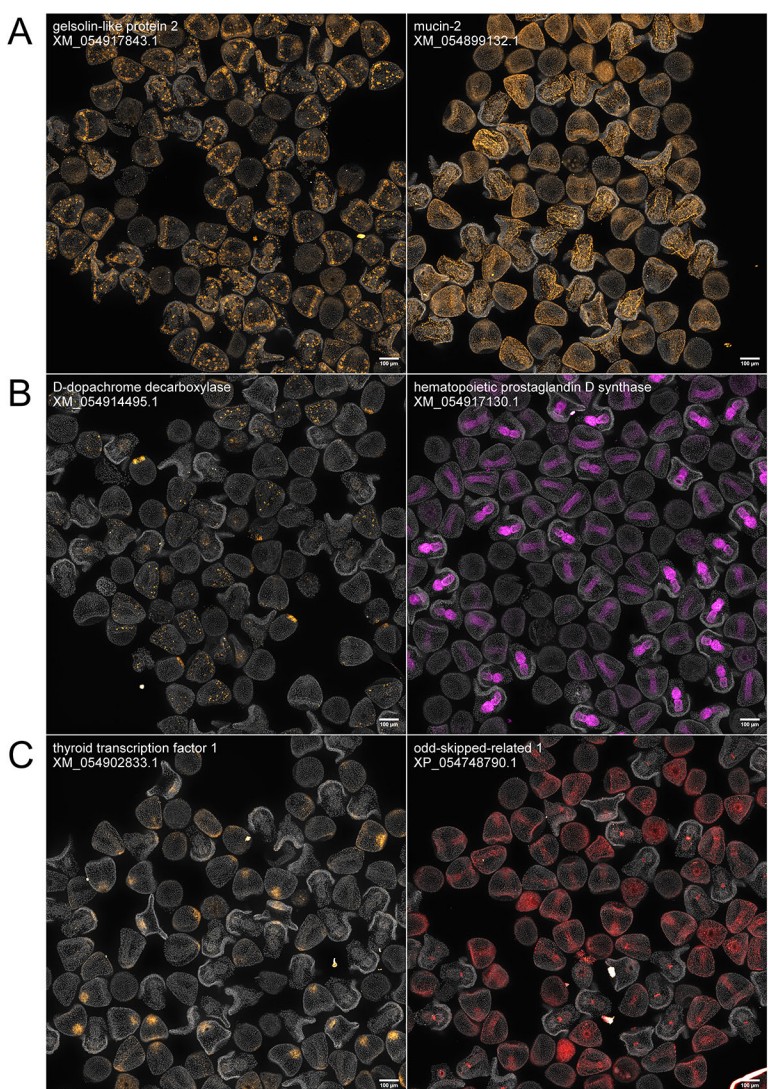

**Fig. 4. HT-HCR to localize previously undescribed cellular structure, enzyme and transcription factor genes.** All images contain a mix of 12, 24 and 36 hpf embryos, and are automatically demultiplexed and stitched outputs resulting from an automated image processing workflow. Each image contains between 50 and 100 embryos, and are annotated with the NCBI RefSeq ID for the mRNA sequence targeted by probe oligonucleotides. (A) Genes involved in cellular structure: *gelsolin-like protein 2* and *mucin-2*. *gelsolin-like protein 2*: expression does not appear to be present in 12 hpf embryos. In 24 hpf embryos, expression is visible in PMCs and SMCs. *mucin-2*: expression is faint or absent in 12 hpf embryos. Expression is visible in the ectoderm in 24 and 36 hpf embryos. (B) Enzymes: *D-dopachrome decarboxylase* and *hematopoietic prostaglandin D synthase*. *D-dopachrome decarboxylase*: expression is visible in the NSM in 12 hpf embryos. Expression is visible in mesodermal cells in 24 hpf embryos. Expression is faint in 36 hpf embryos – some samples show signal in mesodermal cells and the midgut. *hematopoietic prostaglandin D-synthase*: expression is not present in 12 hpf embryos. In 24 hpf embryos, expression is visible in the gut tube. By 36 hpf, expression is localized in the midgut and foregut. (C) Transcription factors: *thyroid transcription factor 1* and *odd-skipped related 1 protein*. *thyroid transcription factor 1*: expression is either faint or absent in 12 hpf embryos. In 24 hpf embryos, expression is visible in the apical ectoderm. In 36 hpf embryos, expression is visible in the oral hood. *odd-skipped related 1 protein*: expression is faint or absent in 12 hpf embryos. In 24 hpf embryos, expression appears to concentrate in the gut tissue. By 36 hpf, expression is restricted to the cardiac sphincter. Scale bars: 100 µm.

by Yaguchi et al. (2016). *homeobrain* is concentrated in the apical ectoderm in 12 hpf embryos (Fig. 3C). At 24 hpf and 36 hpf, expression is in the apical ectoderm and stomodeum. In more developed 36 hpf embryos, expression can be seen in the upper region of the developing oral hood.

Next, we localized several transcription factors whose expression patterns were enriched at highly specific timepoints. These included the expression patterns of *XHOX-3/eve2* (Peter and Davidson, 2010), *deadringer* (*dri*) (Amore et al., 2003) and *Six1* (Byrne et al., 2018) (Fig. 3D-F). *XHOX-3/eve2* expression is restricted to a ring of cells in the aboral ectoderm in 12 hpf embryos (Fig. 3D). In 24 and 36 hpf embryos, expression is faint or absent. *dri* expression is highly concentrated in the oral ectoderm in 24 hpf embryos (Fig. 3E). Expression is absent in 12 hpf embryos; in 36 hpf embryos, some faint expression is visible in the ciliary band. The expression of *Six1* is clearly visible in 36 hpf embryos in one of the coelomic pouches, and in 12 and 24 hpf embryos, the expression pattern is either sparse or absent (Fig. 3F). Overall, these results indicate that HT-HCR is robust across spatiotemporal domains.

### Expression pattern discovery using the automated assay

An application of automated HCR is localization of understudied genes, at reduced cost and effort. Our screen revealed the expression patterns of several genes that, to our knowledge, have not been previously localized in urchins (Fig. 4). These include genes important for cellular structure, such as *gelsolin-2* and *mucin-2* (Fig. 4A). *gelsolin-2* is expressed in mesenchymal cells in 24 and 48 hpf embryos, but expression is absent in 12 hpf embryos. *mucin-2* is expressed in the ectoderm in 24 and 48 hpf embryos, but expression is absent in 12 hpf embryos. We localized the expression patterns of enzymes such as *D-dopachrome decarboxylase* and *HPDS* (Fig. 4B). *D-dopachrome decarboxylase* is expressed in the non-skeletogenic mesoderm (NSM) in 12 hpf embryos. In 24 hpf embryos, expression is localized in mesodermal cells and expression appears to fade in 48 hpf embryos. *HPDS* expression is not visible in 12 hpf embryos, appears to be in the whole gut tube in 24 hpf embryos, and isolated to the mouth and midgut tissues in 36 hpf embryos. New transcription factors such as *thyroid transcription factor-1* and *odd-skipped related-1* were localized (Fig. 4C). Expression of *thyroid transcription factor-1* is absent in 12 hpf embryos and is concentrated in the apical organ in later stages. Expression of *odd-skipped related-1* appears to be weak or absent in 12 hpf embryos. At 24 hpf, expression is visible in the ectoderm; by 48 hpf, expression appears to be localized in the cardiac sphincter.

Next, we also applied this tool to localization categories of genes that are challenging to localize by *in situ* hybridization due to the

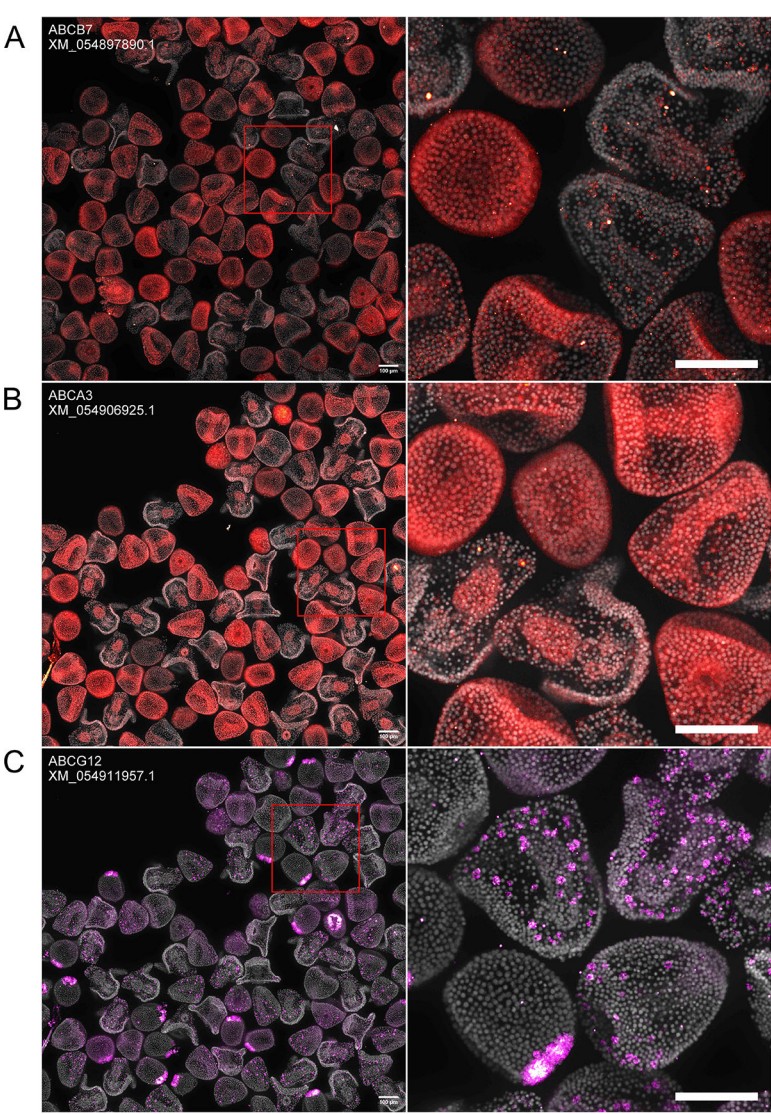

**Fig. 5. Previously unreported ABC transporter expression patterns revealed by HT-HCR.** Images were collected and annotated as in Fig. 4. (A) *ABCB7:* expression is low in 12 hpf embryos. Expression is somewhat ubiquitous but concentrated in mesodermal cells in 24 hpf embryos. Expression is restricted to pigment cells by 36 hpf. (B) *ABCA3:* expression is ubiquitous in 12 hpf and 24 hpf embryos. Expression is concentrated in the midgut in 36 hpf embryos. (C) *ABCG12:* expression is concentrated in the NSM in 12 hpf embryos. At 24 hpf and 36 hpf, expression is concentrated in mesodermal cells. Scale bars: 100 μm.

presence of multiple related homologs and/or relatively low expression levels. Examples were ion channels and transporters, including ATP-binding cassette (ABC) transporters and solute carriers (SLCs), as well as other membrane transporters belonging to additional gene families.

For ABC transporters (Fig. 5A-C), new localizations included heme transporter *ABCB7* (Allikmets et al., 1999; Pondarré et al., 2006; Pondarre et al., 2007; Taketani et al., 2003) and lipid transporter *ABCA3* (Ban et al., 2007; Klugbauer and Hofmann, 1996; Schmitz and Langmann, 2001; Xie et al., 2022). The localization of lipid/sterol transporter *ABCG12* (Chen et al., 2011; McFarlane et al., 2010; Pighin et al., 2004) has previously been demonstrated in *S. purpuratus* (Lee et al., 2023) and was repeated for *L. pictus* in this study. *ABCB7* expression is ubiquitous in 12 hpf embryos. By 24 hpf, expression is still ubiquitous but more concentrated in what appear to be mesodermal cells (based on their location within the embryo), and by 36 hpf, *ABCB7* becomes more consistent with mesodermal cell expression (Fig. 5A). *ABCA3* expression is ubiquitous in both 12 hpf and 24 hpf embryos, but by 36 hpf, the expression is restricted to the midgut (Fig. 5B). The expression pattern for *ABCG12* followed the stereotypical spatiotemporal expression pattern of mesodermal genes (Fig. 5C). 12 hpf embryos showed concentrated expression in the NSM ring,

and by 24 hpf and 36 hpf, expression was concentrated in mesodermal cells.

For SLCs, new localizations included the choline transporter *SLC5A7* (Apparsundaram et al., 2000; Iwamoto et al., 2006; Okuda and Haga, 2000), the vesicular acetylcholine transporter SLC18A3 (Varoqui and Erickson, 1996; Weihe et al., 1996), and the neutral and dibasic amino acid transporter *SLC6A14* (Anderson et al., 2008; Sloan and Mager, 1999) (Fig. 6A-C, Fig. S6). At 12 hpf, the expression of *SLC5A7* is sparse or absent (Fig. 6A). In 24 hpf embryos, expression is restricted to cells that reside in the putative location of post-oral neural progenitor cells described by Slota et al. (2019). In 36 hpf embryos, expression was sparse or absent (Fig. 6A). The expression pattern of *SLC18A3* also resembles those of genes expressed in post-oral neural progenitor cells at 24 hpf (Fig. 6B). By 36 hpf, *SLC18A3*[+] cells are scattered along the ciliary band. *SLC6A14* expression is sparse or absent in 12 hpf and 24 hpf, but by 36 hpf, the expression is restricted to one of the coelomic pouches (Fig. 6C).

We also localized other understudied membrane transporters crucial to embryo survival and growth (Fig. 7). One of these was *aquaporin-8* (Ishibashi et al., 1997; Koyama et al., 1997), a member of the aquaporin superfamily of membrane water channels (Agre and Kozono, 2003; King et al., 2004) that enables a rate of passage

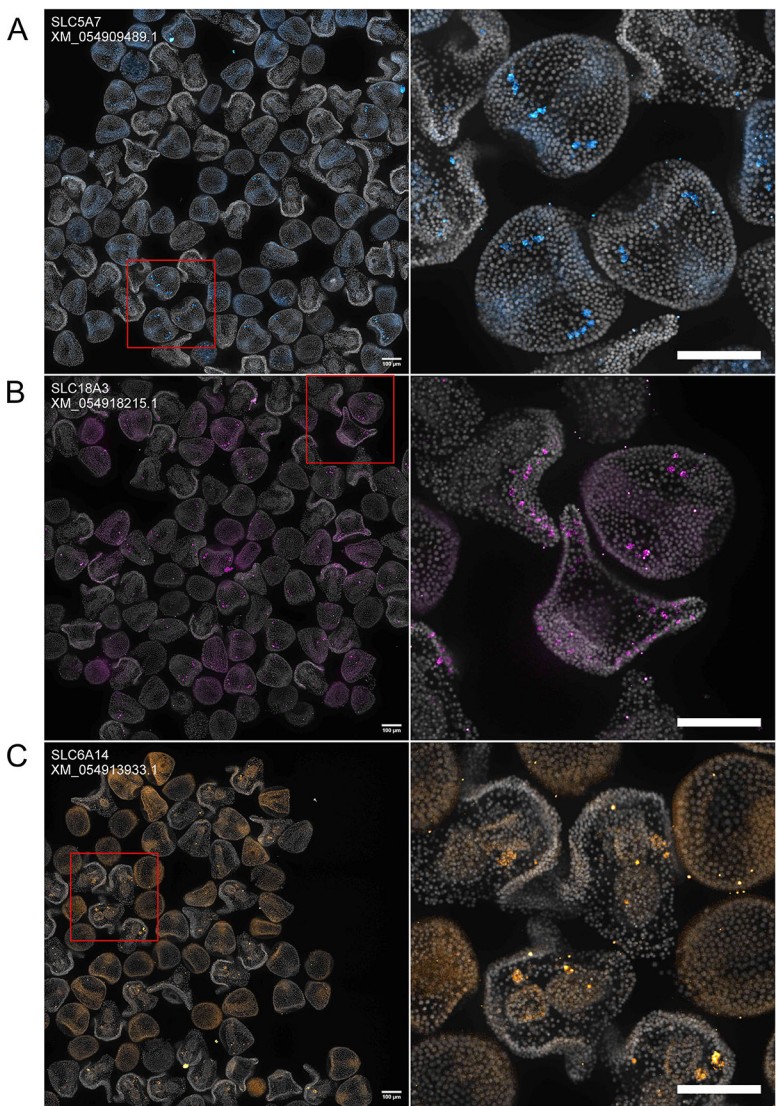

**Fig. 6. Previously unreported SLC transporter expression patterns revealed by HT-HCR.** Images were collected and annotated as in Fig. 4. (A) *SLC5A7*: expression is either sparse or absent in 12 hpf embryos. At 24 hpf, expression is present in what appears to be neuronal precursor cells. At 36 hpf, expression is either sparse or absent. (B) *SLC18A3*: expression is either sparse or absent in 12 hpf embryos. At 24 hpf, expression is concentrated in what appears to be neuronal precursor cells. Expression is concentrated in cells distributed throughout the ciliary band in 36 hpf embryos. (C) *SLC6A14*: expression is either sparse or absent in 12 hpf and 24 hpf embryos. At 36 hpf, expression is concentrated in one of the coelomic pouches. Scale bars: 100 µm.

of water molecules across the cell membrane that is orders of magnitude higher than that of simple diffusion (Kozono et al., 2002; Law and Sansom, 2002). The expression of *aquaporin-8* is absent in 12 hpf embryos (Fig. 7A). At 24 hpf, expression is restricted to the premature hindgut and midgut, and is excluded from the foregut (Fig. 7A). The expression pattern is continued in 36 hpf embryos (Fig. 7A). Finally, another important membrane protein that was localized was *short transient receptor potential channel 4* (*strp4*). *strp4* is expressed in mesodermal cells (Fig. 7B). At 12 hpf, expression is present in the NSM, and at 24 and 36 hpf, expression is concentrated in mesodermal cells.

## DISCUSSION

For four decades, ISH has been a core methodology in the sea urchin community. Yet, while it is difficult to precisely determine the number of genes localized, we estimate the total to still be only several hundred genes. As demonstrated here, robotic liquid handling technologies and HCR have the potential to dramatically accelerate the pace of discovery using this animal model, and likely several others with similar biological features. The localization of 101 genes in *L. pictus* is itself an important foundational step for this animal model, as this species is the first echinoderm in which both

transgenic and knockout lines have been produced (Jackson et al., 2024; Vyas et al., 2022).

## Features and limitations of HT-HCR

HT-HCR is a high-throughput, automated and miniaturized version of the established manual HCR protocol, and thus faces the same limitations related to reagent chemistry and interpretation of image data. We considered multiple factors that might contribute to difficulty in interpretation of signal from a given probe set. In general, *in situ* hybridization is not ideal for the investigation of genes that are not highly expressed and/or concentrated in a specific region (Fig. S2B,C). In our hands, abundant genes were generally more likely to produce robust signal; however, this was not always the case, as in the example of SLC18, which was a relatively rare transcript that produced good signal presumably because it was concentrated in very few cells. For genes with multiple paralogs or isoforms, negative results could be a product of the specific paralogs or annotations chosen for probe design. Indeed, of these 91 probe sets, 15 were designed to target genes with multiple transcript variants. Consistent with this idea, for *mucin-2* and *mucin-17*, two probe sets for these specific transcript variants showed clear expression patterns, while three (two for *mucin-2* and one for *mucin-*

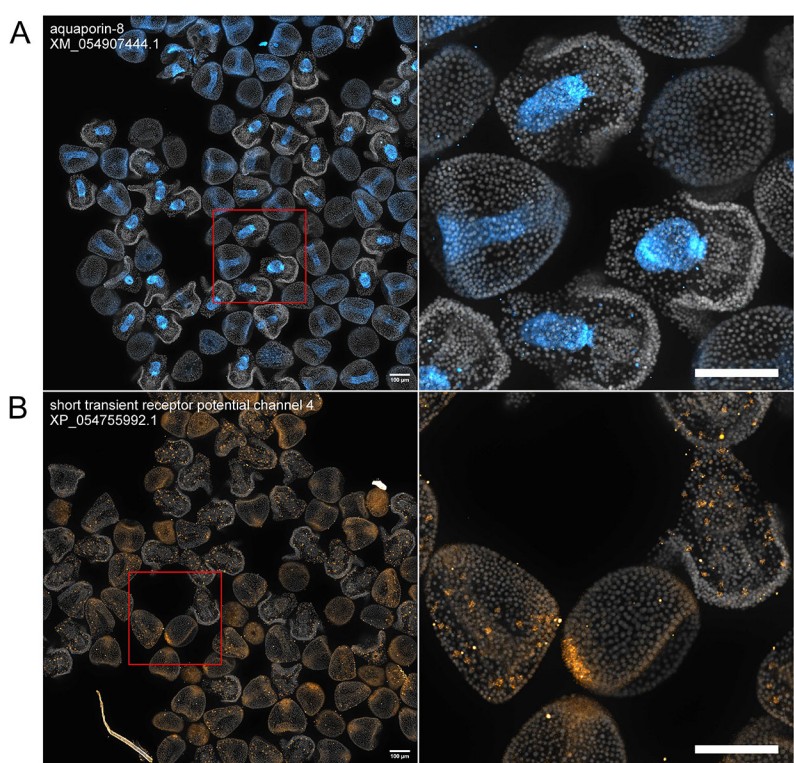

**Fig. 7. Aquaporin-8 and Strp-4 expression patterns revealed by HT-HCR.** Images were collected and annotated as in Fig. 4. (A) *aquaporin-8*: expression is either sparse or absent in 12 hpf embryos. At 24 hpf and 36 hpf, expression concentrated in the gut. (B) *short transient receptor potential channel 4*: at 12 hpf, expression is concentrated in the NSM. At 24 hpf and 36 hpf, expression is concentrated in mesodermal cells. Scale bars: 100 µm.

*17*) did not, and were thus considered part of the 91 negatives. It is also conceivable that some of the probe sets correspond to genes that were simply not expressed strongly enough to detect in one of the stages assayed.

We speculated that manual HCRs might produce better quality images due to concerns involving the deterioration of reagent quality from sitting on the robotic deck at room temperature throughout the assay. However, as shown (Fig. 2), we found no detectable difference in image quality between automatic runs and the manual runs. We were also concerned about whether the automated handling of embryos would harm the morphology of the samples. There was no clear indication of damage to or loss of embryos using automated pipetting, as the robotic protocol was designed to follow the same precautions taken during manual handling of embryo samples.

In this study, the HT-HCR protocol was used to process the commonly assayed developmental stages in the sea urchin research community. We expect that embryos in any developmental stage prior to 36 hpf will be viable for use with this method, as long as the fertilization envelope is removed for pre-hatching stages. However, the current HT-HCR protocol will require modification to handle larger embryos from later developmental stages (>250 µm). This is because the protocol relies on the samples themselves being moved across different vessels throughout the assay, thereby enabling the use of highly miniaturized reaction volumes (and thus reductions in the use of costly reagents, i.e. hairpins and amplifiers). A possible solution to this problem could be to use larger volumes, although this would increase the cost of the assay per sample and would still be limited by the maximum embryo size (~0.5 mm) that can be moved by the handler. Automated protocols for larger samples, such as late-stage larvae and juveniles, are unlikely to achieve the level of efficiency in reagent usage and imaging throughput presented here.

Finally, we also considered that many research projects in the echinoderm research community have been and may continue to be low throughput. Outside of large perturbation screens, the scale of

HT-HCR may not be necessary for many applications. This said, our HT-HCR protocol was written for operation on a common, general-purpose laboratory liquid handler; even small research groups may benefit from automated low- or medium-throughput processing of HCR samples because of the potential to reduce user-to-user variation. In addition the assay may be useful for the validation of single-cell RNA-sequencing datasets. Finally, while our protocol uses a high content confocal imager, any standard confocal microscope could be used if only a small number of samples must be imaged.

### Implications and future directions

This study has several potential future implications for the use of ISH in sea urchins and related echinoderms. The first is that many more genes can now be described, including under-studied gene families. For example, the study of gene regulatory networks (GRNs) in urchins has been driven by extensive localization of the mRNA expression governed by transcription factors (TFs). However, many of the downstream effectors of these regulatory networks, such as the genes encoding structural or physiological proteins, remain under sampled. In addition, using HT-HCR or similar techniques, community-level atlases of gene expression can quickly be acquired and linked to gene annotations (Bell et al., 2004; Telmer et al., 2024; Thisse et al., 2004; Tomancak et al., 2002, 2007), as is the case for other animal models (Bradford et al., 2011; Fisher et al., 2023; Uhlen et al., 2010).

Perhaps more importantly, because HT-HCR results in dramatic reduction in the time and cost per sample, it makes various types of perturbation analyses feasible. By nature of the readout, the results are directly relatable to the morphology of the embryo and, thus, might provide insight into the mode of action of small molecule signals and teratogens. This could be particularly transformative for studying the effects of diverse small molecules on development using wild type, knockout (Tjeerdema et al., 2024) and transgenic urchins. These range from environmental chemicals (Anderson

et al., 1994; Gambardella et al., 2021; Mijangos et al., 2020; Rendell-Bhatti et al., 2021; Sartori et al., 2023) to drugs (Gunaratne et al., 2018; Kim et al., 2022; Semenova et al., 2006; Tjeerdema et al., 2024) to endogenous molecules (Onjiko et al., 2015, 2016) – all of which have been speculated to act on development, but remain challenging to study at scale. To date, perturbation studies using ISH in urchins have necessarily focused on the expression of a limited subset of gene targets (Paganos et al., 2023; Rodríguez-Sastre et al., 2023). This approach opens the door to screening entire libraries of small molecule compounds against numerous genes, and could thus facilitate a new era of chemical biology using the sea urchin embryo.

Animal models have historically been selected because of their unique biological features (Hamdoun et al., 2023) that make them advantageous for certain avenues of investigation. In the case of the sea urchin, the biological features are extreme fecundity and developmental synchrony. This made it the model of choice for 'biochemical scale' dissection of the molecular sequences of events occurring during development (Davidson et al., 2002; Evans et al., 1983). However, this advantage was diluted as the field transitioned from biochemical methods that utilize millions of embryos, to molecular and imaging-based methods that, at most, leveraged 100s of embryos. This study demonstrates how the unique features of the sea urchin may continue to be advantageous for high-throughput, automated analysis of development.

## MATERIALS AND METHODS
### Probe design
Probe sets for 192 targets predicted in the genome assembly UCSD_Lpic_2.1 were designed using a version of insitu_probe_generator (Kuehn et al., 2022) containing minor modifications to facilitate the design of probes in multiplexed oligonucleotide pools, automatic retrieval of target information and metadata from NCBI, input of target coding sequences (CDS), and generation of metadata that tracks and labels intermediate outputs of the probe pool design process (https://github.com/ylee-sio/HL_HTHCR). These were necessary for data management and image processing pipeline.

Individual probes were all designed to have a maximum homopolymer length of three nucleotides. The maximum number of probe pairs (i.e. probe set) for a given target gene was 15. Because the number of probe pairs depends on the length of the target gene, the number of probe pairs per target gene varied from 3 to 15 (Tables S1 and S2). Each probe set was designed to be specific to one of four different amplifier sequences, B1, B2, B3 and B4, to multiplex samples in a probe pool for four gene targets (Choi et al., 2018). Thus, each probe pool contained four probe sets. A total of 48 probe pools were ordered and synthesized (Integrated DNA Technologies, Coralville, Iowa, USA) at a scale of 100 pmol per oligonucleotide sequence. Oligonucleotide pools were resuspended in 100 µl of 1×TE buffer or nuclease-free water to obtain a 1 µM concentration per oligonucleotide per pool and were stored in −20°C.

### Embryo culturing and fixation
Adult *Lytechinus pictus* were collected in La Jolla, CA and kept at 18-22° C in ambient seawater. Animals were spawned by injection with 0.55 M KCl. Three independent batches were generated for each of the 24 hpf embryos and the mixed stages. For each batch of embryos, sperm from one male and eggs from two to four female sea urchins were spawned and mixed according to Nesbit and Hamdoun (2020).

Eggs were washed with 0.2 µm filtered sea water (FSW) and any debris collected while spawning was removed. After fertilizing the eggs, the culture density was normalized to 500 embryos/ml in 2 l beakers. Cultures were raised at 21°C in filtered sea water. Embryos were collected at 12 (blastula), 24 (gastrula) and 36 (pluteus) hours post fertilization (hpf) by aspirating off the surface with a 50 ml serological pipette and concentrating the embryos on a 75 µm strainer. The embryos were then transferred to a 15 ml tube and allowed to settle on ice. Next, the supernatant was removed until 2 ml of embryos in FSW remained. This was transferred to a 2 ml

cryogenic tube, and the concentration of embryos was ~250,000 embryos/ml. Concentrated embryos were placed on ice for 10 min to allow the embryos to settle to the bottom of 2 ml tubes. This step was repeated in between all steps that involve agitation or mixing of the embryos. Once settled, 1 ml of the supernatant was removed, and 1 ml ice-cold 1 M NaCl was added to bring the sample to a final concentration of 0.5 M NaCl to deciliate the embryos. Following deciliation, as much supernatant as possible was first removed, followed by two 1.5 ml washes with FSW. After the final FSW wash, 1 ml of the supernatant was removed and 1 ml of 8% paraformaldehyde was added to 1 ml of the embryos, bringing the final paraformaldehyde concentration to 4%. Samples were incubated at 4°C on a rocker for 48 h. One important deviation to note here from previously published HCR protocols is that all embryo concentration steps were performed by putting embryos on ice to cease their swimming, rather than using centrifugation. The reason for this was that centrifugation inevitably introduced undesirable debris and clumping into the final prepared sample.

After incubation in 4% paraformaldehyde, the samples were kept on ice and washed three times with 1 ml 1×phosphate buffered saline with a final concentration of Tween 20 at 10%, with 10 min incubation time in between all washes. Subsequently, three washes were performed using 1 ml of 50%, 75% and 100% methanol accordingly. After the final methanol wash, each 2 ml sample was topped off to 2 ml with 100% methanol, bringing each sample tube to ~40-60 embryos/µl. Samples were stored at −20°C for use in the automated HCR assay.

### Consumable and reagent preparation
All consumable plates and reagents were prepared according to the final concentrations found in the HCR RNA-FISH protocol for whole-mount sea urchin embryos (Molecular Instruments). All bulk reagents, probe wash buffer, amplification buffer and 5×saline sodium citrate (SSCT) with a final concentration of Tween 20 at 10% were aliquoted into 12-well, 15 ml trough plates. These bulk reagent plates were heat sealed and stored at 4°C. The first well of each 12-well trough plate was left empty to be loaded with rehydrated fixed embryos.

The following describes the sample rehydration process performed immediately before starting all runs. Fixed embryos in methanol were left upright on ice to settle to the bottom of the 2 ml tube. 1.7 ml of methanol were removed from the sample. Five 1.5 ml washes are performed using 5×SSCT at room temperature. For each wash, samples were incubated until embryos settled to the bottom of the tube. After the final wash, leaving a total of 300 µl of embryos in 5×SSCT, 800 µl of warmed (37°C) hybridization buffer is added to the sample. The sample is left to pre-hybridize at 37°C for at least 1 h. Pre-hybridized embryos were transferred to the empty first well of the bulk reagent plate immediately before the run.

Primary stock probe plates were created by mixing 10 µl from a 1 µM stock probe pool solution with 90 µl of hybridization buffer (HB) into each well of a 96-well plate according to a predefined plate map, bringing the probe pool concentration in each well to 100 nM. An intermediate probe plate was then created by mixing 20 µl from the primary stock probe plate and 80 µl of hybridization buffer to obtain 100 µl of a 20 nM probe solution in each well. The contents of the intermediate probe plate were distributed across 20 consumable probe plates in 5 µl aliquots. This consumable plate is then used directly in the automated assay and discarded after use. All data in this publication have been produced using the same batch of probe plates that have been stored in this way for at least 3 months.

The amplifier hairpins used were B1H1/H2-Cyanine7, B2H1/2-Alexa Fluor 647 and B3H1/H2-Alexa Fluor 594 (Molecular Instruments). We synthesized B4H1/H2 amplifier hairpins by conjugating amine-modified oligonucleotides with the B4 sequences with NHS-ester modified ATTO 532 and subsequently purifying the conjugated product using 15% urea-polyacrylamide gel purification (amine-modified oligonucleotide synthesis: Integrated DNA Technologies; fluorophore: ATTO-TEC). The resulting B4H1/2-ATTO 532 amplifier hairpins were normalized to 3 µM. To make a stock hairpin master mix, 100 µl of each amplifier hairpin at 3 µM were incubated at 95°C for 90 s and snap-cooled at room temperature for 1 h minimum. After incubation, all amplifiers are mixed into a single tube. The concentration of each hairpin at this stage is 375 nM. 25 µl of this master mix was then distributed into each well of the first column across four

96-well plates to be used as consumable amplifier plates. All consumable reagent plates were heat sealed and stored at 4°C.

## Assay automation and miniaturization

Automation and miniaturization of HCR resulting in HT-HCR was accomplished using the Opentrons Flex (Opentrons Labworks, Long Island City, New York, USA), a general-purpose laboratory liquid handler. HT-HCR is a miniaturized adaptation of the standard HCR protocol using reaction volumes reduced from 100 µl to 15 µl. The file encoding this protocol, *HL_HTHCR*, including the offsets from calibration for the modules and consumables, is available at https://github.com/ylee-sio/HL_HTHCR. The modules installed for this protocol included two Flex 8-Channel Pipettes (1-50 and 5-1000 µl), Flex Gripper (robotic arm), Opentrons Thermocycler GEN2, Heater-Shaker Module (with the Universal Flat Adapter), Temperature Module (with the PCR Plate Adapter) and the Flex Waste Chute. Robotic calibration was performed for each module and for the dimensions of the following consumables: full skirted 96-well PCR Plate (ThermoFisher Scientific, AB-3396), 96-deep well plate (Agilent, 5043-9300), 96-well glass-bottom imaging plate (Cellvis, P96-1.5H-N) and 12-well/trough 15 ml reservoir (NEST, 360102).

A consumable probe plate was removed from −20°C, centrifuged, unsealed and loaded onto the thermocycler module set at 37°C. The consumable amplifier plate was removed from 4°C, centrifuged, unsealed and loaded onto the temperature module at 12°C. The temperature module was set to be 1-2°C above the temperature of the local dew point to prevent condensation on the bottom of the module. 1 ml of prehybridized embryos was dispensed into the empty trough left for loading fixed samples. The bulk reagent plate was then loaded onto the deck at room temperature.

Using the 8-channel pipette module, 10 µl of embryos were transferred from the bulk reagent plate to the consumable probe plate on the thermocycler and mixed vigorously, bringing the final concentration of each individual probe to 6.66 nM in a 15 ml final volume in each well. Following this, the thermocycler was set to close and seal the plate with the lid set at 40°C. This temperature was set to avoid condensation on the thermocycler seal. The embryos in the consumable probe plate were left to hybridize for 12 h. After hybridization, samples were washed three times using 135 µl of probe wash buffer at 37°C, with a 30 min settling period following each wash. Following this, samples were washed three times using 135 µl of 5×SSCT at 21°C, with a 30 min settling period following each wash. After the final 5×SSCT, the plate contained 15 µl 5×SSCT with embryos settled at the bottom of each well. Samples were resuspended using 30 µl of amplification buffer. Before amplification, 35 µl of samples suspended in amplification buffer were moved to the unused columns of the consumable amplifier plate, bringing the volume of samples in the probe plate down to 10 µl.

70 µl of amplification buffer was added to each well of the 25 µl master mix, resulting in a concentration of 98 nM per probe. 12 µl of this diluted hairpin solution was transferred to the samples in the consumable probe plate, resulting in a final concentration of 53 nM per hairpin. Samples were incubated in hairpin solution for 12 h at 21°C. After amplification, samples were washed using 135 µl 5×SSCT at 21°C, leaving 30 min between each wash for embryos to settle. Samples were then incubated in 135 µl 5×SSCT containing Hoechst 33342 for nuclear staining at 10 µg/ml at 21°C for 30 min. This nuclear stain in 5×SSCT was then removed and a 135 µl 5×SSCT was added to resuspend the embryos. Immediately after resuspension, 135 µl of the solution containing the resuspended embryos were transferred to a glass-bottom imaging plate sitting on a shaker module. After completing all transfers to the imaging plate, embryos were centered in their wells by shaking the plate at 500 rpm for 5 min to concentrate embryos in the center of the well and prepare them for imaging.

## Imaging

The imaging plate was loaded into the ImageXpress HT.ai Confocal Microscope (IXM HT.ai; Molecular Devices). Samples were imaged with a 20×0.8NA Plan Apo Lambda objective (Nikon) with the following configurations for the laser light source (89North, Williston, Vermont,

USA) and emission filters (Semrock, Rochester, New York, USA) combinations: 740 nm with a 794/32 filter for Cyanine7, 640 nm with a 680/42 filter for Alexa Fluor 647, 555 nm with a 624/40 filter for Alexa Fluor 594, 520 nm with a 562/40 filter for ATTO 532 and 405 nm with a 452/45 filter for Hoechst 33342. Exposure times were set for 150 ms for all channels except for the 405 nm channel, which was set at 50 ms due to the higher intensity of the nuclear stain signal.

For a standard circular bottom image plate, there were a total of 81 sites (9 by 9 grid) available to image using a 20×objective. However, only the center 4-9 sites (2×2 or 3×3) were imaged for data storage efficiency. For images with individual stages of embryos, z-slices were taken at 2 µm intervals. Thirty-five slices were taken for 12 hpf samples, 40 slices were taken for 24 hpf samples and 55 slices were taken for 36 hpf samples. This covers 70 µm, 80 µm and 110 µm for 12, 24 and 36 hpf samples, respectively. The total duration for imaging 48 wells of samples ranged between 3-4 h depending on the developmental stage of the sample. A total of 75,600 (151.2GB), 86,400 (172.8GB) and 118,800 (237.6GB) images were generated for 12 hpf, 24 hpf and 36 hpf runs, respectively (# sites×# slices×# wells×# channels), for nine (3×3) sites. An example of a single site image from a plate, and a schematic of developmental stages imaged in this study are shown in Fig. 1.

In order to reduce assay run times, reagents, consumables, data storage and image processing time, we opted for performing the assay for developmental stage comparisons using mixed stage samples. For assay runs containing mixed stage samples, equal parts of fixed embryos from the three different developmental stages were mixed at the rehydration step. No other changes were made for running the assay for mixed stage samples. For images containing mixed developmental stages, 45 z-slices were taken in 2 µm intervals, resulting in 97,200 images (194.4 GB). All imaging was performed at room temperature (20-23°C).

## Data management and processing

Custom scripts for automatic offloading and archival management of data from the IXM HT.ai acquisition software (MetaXpress 6 Software) were written in bash, Microsoft Powershell and Python. Scripts for automated metadata annotation and application (target text annotation on images, pseudocolor assignment, channel multiplexing and demultiplexing, aspect ratio adjustment according hardware binning levels, site tiling, and directory management tasks) were written in bash, R and ImageJ Macros (Schneider et al., 2012). These scripts are available at https://github.com/ylee-sio/HL_HTHCR.

After data acquisition of all samples, images were automatically sorted by well ID (which links to an oligo pool ID), site ID and wavelength (which links to the hairpin amplifier sequence and fluorophore used for the localization of a specific gene). The 45 images of each z-slice from each site for each wavelength were maximally projected and the center site image, which typically contained the greatest number of embryos within a site among all sites, was used to determine whether the probe produced a positive signal. Generally, we selected images with signal that was visually discernible from background (signal from non-sample area). For presentation of images from 101 probes with positive localization results, images were either stitched to form either the full 3×3 or 2×2 tiled images, or used as single sites. Information linked to well IDs and wavelengths were used to automatically annotate each image with a common gene name and NCBI accession number for the specific mRNA sequence used to design the probe set.

## Acknowledgements
We thank Dr Victor Vacquier for reading and commenting on the manuscript.

## Competing interests
The authors declare no competing or financial interests.

## Author contributions
Conceptualization: Y.L., A.H.; Data curation: Y.L., A.H.; Formal analysis: Y.L., R.M., G.R., E.T., E.W.J.; Funding acquisition: A.H.; Investigation: S.K., E.T.; Methodology: Y.L., C.J., S.K., E.T.; Project administration: Y.L., A.H.; Software: Y.L.; Validation: E.T.; Visualization: Y.L.; Writing – original draft: Y.L., A.H.; Writing – review & editing: Y.L., C.J., E.T., E.W.J., A.H.

## Funding

This work was supported by the National Institutes of Health (ES035541), the National Science Foundation (2414798), the Allen Discovery Center Program (a Paul G. Allen Frontiers Group advised program of the Allen Family Philanthropies), and the Illumina Foundation. Open Access funding provided by the University of California. Deposited in PMC for immediate release.

## Data and resource availability

Scripts for generating probes and protocol file for HL_HTHCRv16 are available at https://github.com/ylee-sio/HL_HTHCR. Images for the 101 localizations can be viewed and downloaded at Hamdoun Lab Image Repository. All other relevant data and details of resources can be found within the article and its supplementary information.

## Peer review history

The peer review history is available online at https://journals.biologists.com/dev/lookup/doi/10.1242/dev.204814.reviewer-comments.pdf

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
