## [Peer Review File · Development (Cambridge, England)]

Automated, high-throughput in-situ hybridization of sea urchin (*Lytechinus pictus*) embryos

Yoon Lee, Chloe Jenniches, Rachel Metry, Gloria Renaudin, Svenja Kling, Evan Tjeerdema, Elliot W. Jackson and Amro Hamdoun
DOI: 10.1242/dev.204814

Editor: Swathi Arur

Review timeline

Original submission:	26 March 2025
Editorial decision:	16 June 2025
First revision received:	5 August 2025
Accepted:	20 August 2025

Original submission

First decision letter

MS ID#: dev.204814

MS TITLE: Automated, high-throughput in-situ hybridization of *Lytechinus pictus* embryos

AUTHORS: Yoon Lee, Chloe Jenniches, Rachel Metry, Gloria Renaudin, Svenja Kling, Evan Tjeerdema, Elliot W. Jackson and Amro Hamdoun

Dear Dr Hamdoun,

I have now received all the referees' reports on the above manuscript, and have reached a decision. The referees' comments are appended below, or you can access them online: please go to:

As you will see, the referees express considerable interest in your work, and also recommend suggestions that will improve the rigor of the work, and clarity. If you are able to revise the manuscript along the lines suggested, which may involve further experiments, I will be happy receive a revised version of the manuscript. Your revised paper will be re-reviewed by one or more of the original referees, and acceptance of your manuscript will depend on your addressing satisfactorily the reviewers' major concerns. Please also note that Development will normally permit only one round of major revision.

Please attend to all of the reviewers' comments and ensure that you clearly highlight all changes made in the revised manuscript. Please avoid using 'Tracked changes' in Word files as these are lost in PDF conversion. I should be grateful if you would also provide a point-by-point response detailing how you have dealt with the points raised by the reviewers in the 'Response to Reviewers' box. If you do not agree with any of their criticisms or suggestions please explain clearly why this is so.

Reviewer 1

Advance summary and potential significance to field

In this manuscript, Lee et al. present a new protocol for high-throughput HCR in situ hybridizations on sea urchin embryos, using a generic liquid handling machine to automatize this assay. This is a

valuable technique that uses a multipurpose lab equipment, rather than requiring the need to invest into existing in situ robots that are by design limited to a single purpose. The protocol, which includes both the wet lab part of the HCR assays and the downstream data management process is well detailed. The quality of the data produced using this method is impressive and is on par with manual HCRs. This technique will be valuable to increase the throughput of gene expression surveys, in particular downstream of perturbation experiments, or to screen large numbers of genes such as structural genes that are rarely surveyed by in situ hybridizations. Because the use of HCRs is rapidly growing in the developmental biology community, the automation protocol presented in this manuscript is an interesting resource, which I recommend for publication. This being said, I have a few minor comments that the authors should address to improve the quality of the manuscript.

- The quality of the data is impressive and really speaks for itself, therefore it is unfortunate that at times it feels that the authors are unnecessarily overselling their method. For instance, they state "The advent of stable genetics in the sea urchin *Lytechinus pictus* (Jackson et al., 2024; Vyas et al., 2022) has paved the way for on-demand generation of millions of mutant". I appreciate the enthusiasm of the authors and the development of stable transgenesis in *L. pictus* is a formidable achievement, but millions seems a bit overblown here. Later, the authors state "Our approach relies on plate and robotic formats likely to be found in many core facilities or individual lab". In my experience this kind of equipment is not standard in most marine stations (outside of Scripps or MBL maybe), even though marine stations are the most likely places where this protocol will be of interest given that it is tailored for sea urchins embryos. Without undermining the value of the protocol presented here, it would be fair to say that the material required to implement this technique will likely be a limit to its broad use by the community.

- In the same vein, I am being circumspect as to whether in situ hybridizations qualify as "spatial transcriptomics", as stated by the authors, and would rather avoid using this term just for the sake of riding on the latest trend of the community.

- The protocol presented here is tailored for early developmental stages of sea urchins embryos. I am wondering how easy it would be to adjust the protocol for other species or other types of samples, and it would be nice if the authors could elaborate a little bit more on this point. Have the authors tried embryos from other species? There are many other marine organisms available at Scripps that would likely only require minimal adjustments to the protocol. This would help reach a much broader audience.

- It would be valuable for the code of the modified version of the in situ probe generator to be made available.

- I noticed that the protocol include a deciliation step, and I was curious to know the rationale for including this step. Does the presence of cilia interfere with downstream HCRs? It would be interesting to write down the reason for this step in the protocol.

- Rather than the experiment itself, one bottleneck of HCRs is the time required for imaging. Here, the authors use an ImageXpress HT.ai Confocal Microscope. I am not familiar with this technology, but it seems to be tailored for high-throughput and plate imaging. I am curious if the protocol is versatile enough so that it can be implemented with more broadly used confocal microscope options such as Leica or Zeiss. It would be good for the authors to provide more details about this point, and state what are the specs required for an imaging system to be able to implement this protocol.

- In the methods, the second paragraph of the imaging sections was not very clear for me when explaining how the 96 well plates were imaged. I get that the sites correspond to the tiles of a larger tiling grid that were imaged for each well, but this should be explained more clearly (maybe a schematic showing a plate with what exactly is being imaged could help).

- Throughout the figures, it would be good to label the different stages and orientation of the samples described in corresponding section of the results, in particular for readers not familiar with sea urchin embryos.

- For figure 4, it seems that an actual description of these new expression patterns is missing in the results.

Reviewer 2

Advance summary and potential significance to field

Lee et al here combine the strengths of hybridization chain reaction in situ hybridization (HCR) with the synchronization and large number of embryos produced by sea urchins to perform a highly parallel survey of gene expression. Boosted by the utilization of robotic buffer exchange and automated confocal imaging, the authors show over one-hundred gene expression patterns across 3 stages of embryogenesis. While there have been previous high throughput HCR methods published (Gandin et al. 2025 Science) the High-Throughput HCR (HT-HCR) method presented here does not try to probe all genes in a single embryo, but rather, utilizes the enormous number of embryos produced from a single spawning of their model to run many minimally multiplexed reactions simultaneously. This parallelization allowed the authors to use standard reagents without barcoding interpretation as in MerFISH, SeqFISH, or Barcoded HCR. The authors demonstrate that the HCR patterns are more-or-less equivalent when done using automated liquid handling versus when done manually. As part of an attempt to reduce the volume of data collected, the authors used pools of mixed stage embryos that were easily distinguished by morphological characteristics. They then show that for *onecut2*, *scratch*, *homeobrain*, *XHOX-3/eve2*, *Deadringer*, and *Six1* (known transcription factors with distinctive patterns), HT-HCR produces patterns recognizable and comparable to other urchin species. Satisfied with the validation of their method, the authors then move on to use HT-HCR to explore uncharacterized/under-characterized transcripts, as well as closely related paralogs of ABC transporters and solute carrier proteins. On the whole, this report provides a benefit for the urchin community, as well as for other models that produce copious numbers of synchronized embryos and larvae, and will provide a methodology that improves a more wholistic transcriptional understanding of how these organisms respond to perturbation. The authors have optimized the mechanization and automation of HCR while also considerably reducing the working volumes required to perform the reactions (15uL vs 100s). It is also quite exciting that the authors have successfully synthesized their own HCR hairpins. We have a few suggestions below for the manuscript to be considered for publication.

Comments for the author

Major points

Comment 1a: It is a bit surprising to see a roughly 50% failure rate of HCR, 101 successful of 192 probe sets. The authors mention that some of these unsuccessful reactions were among some isoforms of genes probed successfully, were there any other commonalities between failed probe sets? For example, was it correlated with the number of probe pairs, the hairpin used to detect the probes with, did some fail across all channels of the particular reaction (i.e. not necessarily because the probes were poor), and if one extended the amplification step or increased the probe concentration does this improve the results? We believe it would be a service to the readers to include a deeper discussion of the reasons for this rate of failure since this is a resource manuscript.

Comment 1b: In the methods - probe design, the authors indicate that 3-15 pairs of probes were used depending on the transcript length. 3 probe pairs sound very little, but if this consistently works (or fails), it would be great to know, a huge benefit for other HCR users. Tables S1 and S2 can be modified to include probe pair numbers per given gene.

Comment 1c: We also want to invite the authors to submit their probe pool sequences for the sake of reproducibility. Again, especially, failure rate being 50%, this could save others time and resources.

Comment 2: One of the major benefits of HCR is the ability to easily detect 3 or 4 transcripts simultaneously. While the authors show many compelling images that staining works, when presenting data on claimed differentiation of paralogs the images presented are never shown together in a single animal, nor are we shown diagrammatically how probes are designed to differentiate the paralogs (i.e. when sequences are similar, what steps did the authors take to ensure there was no cross reactivity between paralogous mRNAs?). Again, including details on these design decisions would greatly improve the service to the community.

Comment 3: Finally, while the authors include a nice negative control performed by treating the embryos with RNase before probing, suggesting that there was no priming of hairpins off of non-RNA moieties in the embryos, we would like to see a no-probe control done for each of the hairpins used to ensure that there is no non-specific amplification off of an endogenous RNA.

Minor Points

A benefit of this particular technique over other high-throughput ISH methods is that we are shown many embryos simultaneously with similar staining patterns. We appreciate this transparency of

being able to see more than a "representative example", but given the mix of stages and automated nature of the imaging, it is not always easy to evaluate the patterns shown. I believe this paper would benefit from an additional panel with oriented individual embryos for the three timepoints imaged.

In discussion, we want to encourage the authors to point out the use of this high throughput method for single cell RNAseq data validation, when sometimes 100s of transcripts need to be analyzed. This method would make scRNAseq validation much less laborious.

Reviewer 3

Advance summary and potential significance to field

This manuscript describes an automated and miniaturized pipeline for automated HCR FISH that conducts and images sea urchin embryo using robotics and a high-content imager. This represents an important technical advance that opens the door for higher throughput FISH analysis in this model organism. In addition, this study reveals new gene expression patterns for numerous genes in this model organism. Strengths include validations of both reliability, via comparison of this approach and manual FISH analysis for 27 genes, and reproducibility, via explicit depiction of replicates in bulk.

Comments for the author

However, there are also some issues, and the manuscript could be improved as follows.

Major issues:

1. Readers must view and personally interpret each of the FISH images shown, which each contain numerous embryos of mixed stages in random orientations. I think it would be a benefit to the reader to include an additional final figure that schematizes the overall findings, especially for novel gene expression profiles found in Figures 3-7. Embryo schematics for each timepoint with expression patterns painted on them would be ideal. These could potentially instead be added on a per-figure basis rather than collected in a final summary figure, and I'll leave it to the authors to choose how to approach this. This request applies to novel gene expression information; it does not seem necessary to me to also include spatial expression patterns that are already established.
2. In a related manner, Table S1 includes all the genes assessed, but lacks their results. I think it would be the best way to disclose the spatial outcomes of all the FISH experiments in the manuscript by listing the spatial patterns observed at each of the three assessed timepoints by adding an additional three columns to this table.
3. I appreciate that it is lab jargon to call this approach "HCR", but this is vague and ambiguous. HCR stands for hybridization chain reaction (only), as we all know. There is an HCR approach for immunofluorescence labeling as well as one for FISH, both produced by Molecular Instruments. Just saying "HCR" is therefore ambiguous. It's fine for casual talk, but for a publication, precise language is needed. Therefore, each instance of "HCR" should be extended to HCR-FISH. That includes the technique's name, which should be amended to HT-HCR-FISH.
4. The methods section is missing some key information, including the species and source of animals used for the experiments shown and their husbandry, as well as the methods employed for manual FISH. Please explain the rationale both for the repeated cold shocks that the embryos are exposed to prior to fixation and for their deciliation prior to fixation. Each of these strikes me as unnecessary stresses to the embryos that could potentially influence their gene expression, particularly for stress-responsive genes. I see what was written about concentrating this way, but it is unclear to me why concentration by gravity (not literally by "ice", as written) was not performed following fixation to avoid the cold shock. I infer that icing the embryos blocks their swimming, causing them to settle, but this is not clearly explained and should be. The stages employed are described by their time post-fertilization; please also indicate what stage these timepoints correspond to.

5. Regarding those genes that did not generate an interpretable signal, more consideration is needed, and this should be an aspect of the main results section rather than being shunted to supplemental figures and and the discussion. Since ~ half the genes failed at this step, it seems that this method could be seen as half-successful by some readers; thus, an effort to understand why so many do not succeed is warranted.

5a. Within the methods, please clarify the actual probe concentrations employed- are these uniform across all the probes? Was any testing done or were calculation performed to optimize probe concentrations? This is potentially relevant to those probes with negative results. In our hands, HCR-FISH is highly sensitive to probe dose, and this is only logical, since mRNAs are present across a very wide range of concentrations from gene to gene.

5b. Please include plots that compare the expression level at the stages of interest for the successful genes versus the unsuccessful ones to assess whether a higher dose could make a difference.

This point should be made in the results, the plot shown, and this point should also appear in the relevant part of the discussion.

5c. I would like to see a small set (~ 10) of those negative genes re-assayed with a higher concentration of the probes in a dose-response experiment that tests double, triple and quadruple the normal probe dose. I would select them over a range of gene expression levels rather than choosing a group with uniform levels of expression. Please explicitly clarify what these actual probe concentrations are and include this in a main figure.

6. The images from this work were collected, but no further analysis was performed. Please discuss this next component: how will quantification of these data be approached as the next necessary step?

Minor issues

1. The citations in the first paragraph of the introduction do not represent the use of FISH in sea urchins. Only five papers are cited, all more than 10 years old, and only two, from more than 20 years ago, are offered as representative of "long history" of FISH in the field. The remaining three are focused very narrowly and are therefore not representative. The number of cited papers here should be increased to better represent the broad use of FISH in sea urchin developmental biology and should include recent as well as older papers.

2. Figure 1's legend states that five sets of probes are used for each sample, but I think this should instead be four.

3. The first two sentences of the legend for Figures 4-7 are exactly the same. Please revise: The legend for Fig. 5-7 can say that the images were collected and annotated as in Fig. 4 (or similar).

4. There is a lot of statement of results in the legends text that seems more appropriate for the main text or otherwise redundant to it. The legends can thus be more succinct.

5. Supp Figure 2's legend states that tubulin alpha 1 is ubiquitously expressed, but the image shows clear enrichment in what seems to be the PMCs, the ciliary band, and the apical plate at the various timepoints shown.

First revision

Author response to reviewers' comments

SUMMARY

We thank all three reviewers for providing thoughtful critiques of our manuscript, and for noting the value of the technique and the quality of the data. Reviewers 1 and 2 commented that "*the quality of the data is impressive and speaks for itself*" and cited "*benefit for the urchin community, as well as for other models that produce copious numbers of synchronized embryos and larvae*" as well as the potential to provide "*a more holistic transcriptional*

understanding of how these organisms respond to perturbation". Reviewer 3 noted that this is "an important technical advance" with "strengths [that] include validations of both reliability ... and reproducibility". Specific suggestions were provided to improve the manuscript. We considered and addressed each of these carefully, including providing additional data and figures (1C, S3 and S6) as well as updates to existing figures (S4 and Table S1) and the manuscript itself. Our specific responses are below.

REVIEWER 1.

1. The quality of the data is impressive and really speaks for itself, therefore it is unfortunate that at times it feels that the authors are unnecessarily overselling their method. For instance, they state a) "The advent of stable genetics in the sea urchin *Lytechinus pictus* (Jackson et al., 2024; Vyas et al., 2022) has paved the way for on-demand generation of millions of mutant [...]". I appreciate the enthusiasm of the authors and the development of stable transgenesis in *L. pictus* is a formidable achievement, but millions seems a bit overblown here. Later, the authors state b) "Our approach relies on plate and robotic formats likely to be found in many core facilities or individual lab". In my experience this kind of equipment is not standard in most marine stations (outside of Scripps or MBL maybe), even though marine stations are the most likely places where this protocol will be of interest given that it is tailored for sea urchin embryos. Without undermining the value of the protocol presented here, it would be fair to say that the material required to implement this technique will likely be a limit to its broad use by the community.

Thank you for noting the quality of the data! There seems to be a misunderstanding, we were referring to the numbers of embryos produced by urchins (each female can shed >1 million eggs per spawn), not the number of lines. Nonetheless, we removed that paragraph.

Many existing sea urchin labs are at well-equipped major research institutions, or their affiliated marine labs, and liquid handlers are common. This paper reports the second version of the assay we created, and it was specifically designed around a very common, affordable and general-purpose liquid handler (rather than more specialized instruments used in V1). We acknowledge that many labs may still be reluctant to automate HCR and discuss pros and cons of automation for low throughput work. That said, we hope this study will encourage critical reevaluation of the current labor-intensive strategies.

2. In the same vein, I am being circumspect as to whether in situ hybridizations qualify as "spatial transcriptomics", as stated by the authors, and would rather avoid using this term just for the sake of riding on the latest trend of the community.

It seems that, by definition, a technique that maps mRNAs in space and time is a spatial transcriptomic method. In any case, we removed it.

3. The protocol presented here is tailored for early developmental stages of sea urchin embryos. I am wondering how easy it would be to adjust the protocol for other species or other types of samples, and it would be nice if the authors could elaborate a little bit more on this point. Have the authors tried embryos from other species? There are many other marine organisms available at Scripps that would likely only require minimal adjustments to the protocol. This would help reach a much broader audience.

Good point. Since a liquid handler is just an automated way of doing steps otherwise performed manually, this can in principle be used for any embryo that can be moved by the liquid handler. Unfortunately, optimization for other species is beyond what we have resources to take on now.

4. It would be valuable for the code of the modified version of the in situ probe generator to be made available.

All code is available on github and the link is provided under Methods, Assay automation and miniaturization. Unfortunately, it appears that something involved

in the text processing at the journal that damages some of the hyperlinks. We will alert the journal of this. The code can be found here: https://github.com/ylee-sio/HL_HTHCR

5. I noticed that the protocol include a deciliation step, and I was curious to know the rational for including this step. Does the presence of cilia interferes with downstream HCRs? It would be interesting to write down the reason for this step in the protocol.

We followed the original protocol for performing in situ HCR in sea urchins (from Molecular Instruments), which calls for a deciliation step. This protocol can be found here: <https://files.molecularinstruments.com/MI-Protocol-RNAFISH-SeaUrchin-Rev10.pdf>

6. Rather than the experiment itself, one bottleneck of HCRs is the time required for imaging. Here, the authors use an ImageXpress HT.ai Confocal Microscope. I am not familiar with this technology, but it seems to be tailored for high-throughput and plate imaging. I am curious if the protocol is versatile enough so that it can be implemented with more broadly used confocal microscope options such as Leica or Zeiss. It would be good for the authors to provide more details about this point, and state what are the specs required for an imaging system to be able to implement this protocol.

We now have a sentence on this in the discussion. The ImageXpress is just a spinning disk confocal optimized for microwell plates. There are no unique specifications of an imaging system for HT-HCR. Labeled embryos can be imaged with any microscope with appropriate filters/lasers, and plate adapters can be purchased for most confocal and widefield microscopes. The specific software requirements will differ greatly from scope to scope. Users may also still want to automate HCR but image with microscope slides for lower throughput work.

7. In the methods, the second paragraph of the imaging sections was not very clear for me when explaining how the 96 well plates were imaged. I get that the sites correspond to the tiles of a larger tilescan grid that were imaged for each well, but this should be explained more clearly (maybe a schematic showing a plate with what exactly is being imaged could help).

Figure 1B provides a schematic showing how plates were imaged. We now refer to this.

8. Throughout the figures, it would be good to label the different stages and orientation of the samples described in corresponding section of the results, in particular for readers not familiar with sea urchin embryos.

Thank you for this suggestion. We have now included an illustration (Figure 1C) which can be used as a “key” to orient readers.

9. For figure 4, it seems that an actual description of these new expression pattern is missing in the results.

Thanks for catching that. We describe these expression patterns in the text.

REVIEWER 2.

1. Comment 1a: It is a bit surprising to see a roughly 50% failure rate of HCR, 101 successful of 192 probe sets. The authors mention that some of these unsuccessful reactions were among some isoforms of genes probed successfully, were there any other commonalities between failed probe sets? For example, was it correlated with the number of probe pairs, the hairpin used to detect the probes with, did some fail across all channels of the particular reaction (i.e. not necessarily because the probes were poor), and if one extended the amplification step or increased the probe concentration does this improve the results? We believe it would be a service to the readers to include a deeper discussion of the reasons for this rate of failure since this is a resource manuscript.

This is now further discussed in the revised manuscript. There are many reasons an in situ might not produce a clearly interpretable result, and this is a common outcome with in situ screens (Bell et al., 2004; Thisse et al., 2004; Tomancak et al., 2002; Tomancak et al., 2007). In general, the assay reliably localized numerous known control genes, indicating that the method is robust. We also targeted less-studied genes, such as transporters and ion channels, which can be expressed at low levels. Even so, several rare transcripts (e.g., SLC18) were detected, presumably due to high local concentration in a small number of cells. Ultimately, the decision to pursue such challenging targets depends on the needs of the end user. A key application of this automated assay is in de-risking difficult targets, early in a screening pipeline.

In terms of technical factors, there were no “commonalities” to neatly explain localization success. Neither choice of hairpins nor probe numbers appeared to be issues, as we observed successful HCRs across all hairpin types and numbers of probes (Table S1, S2). We also tested the effect of probe concentration and provide additional data in this revision. The standard assay already uses high probe concentrations, and strong signals are detectable even at 10x lower levels (Figure S3). Increasing probe concentration for several weakly expressed genes did not improve their localization, presumably because probe is in molar excess to target.

Finally, while some probes produced weak signals, we preferred not to call a result if we were uncertain of the localization. It is entirely conceivable that some of these would produce acceptable results with further optimization. However, gene specific optimization can involve many conditions, and is somewhat contrary to a high throughput screen.

2. In the methods - probe design, the authors indicate that 3-15 pairs of probes were used depending on the transcript length. 3 probe pairs sound very little, but if this consistently works (or fails), it would be great to know, a huge benefit for other HCR users. Tables S1 and S2 can be modified to include probe pair numbers per given gene.

We have now included the number of probe pairs used for each gene in Tables S1 and S2. We did not see a relationship between number of probe pairs and success. Probe number is simply dictated by gene length; we detected genes with 3 (metallothionein A), 4 (calmodulin) and 5 (P19) probe pairs.

3. We also want to invite the authors to submit their probe pool sequences for the sake of reproducibility. Again, especially, failure rate being 50%, this could save others time and resources.

We provide the exact NCBI accession ID for each target so that anyone can reproduce the sequence. Using the parameters as stated in the methods section, the probe sequences generated for a given target localized in this study will be identical to the sequences used in this study.

4. One of the major benefits of HCR is the ability to easily detect 3 or 4 transcripts simultaneously. While the authors show many compelling images that staining works, when presenting data on claimed differentiation of paralogs the images presented are never shown together in a single animal, nor are we shown diagrammatically how probes are designed to differentiate the paralogs (i.e. when sequences are similar, what steps did the authors take to ensure there was no cross reactivity between paralogous mRNAs?). Again, including details on these design decisions would greatly improve the service to the community.

Probes against paralogous mRNAs were simply designed as any other set of probes. They were designed as probes in different pools, and no pool contained probes for targets which are isoforms to each other, as the goal was to find out which probes for a given transcript isoform produces signal.

5. Finally, while the authors include a nice negative control performed by treating the embryos with RNase before probing, suggesting that there was no priming of hairpins off of non-RNA moieties in the embryos, we would like to see a no-probe control done for each of the hairpins used to ensure that there is no non-specific amplification off of an endogenous RNA.

It has been added to Figure S3.

6. A benefit of this particular technique over other high-throughput ISH methods is that we are shown many embryos simultaneously with similar staining patterns. We appreciate this transparency of being able to see more than a "representative example", but given the mix of stages and automated nature of the imaging, it is not always easy to evaluate the patterns shown. I believe this paper would benefit from an additional panel with oriented individual embryos for the three timepoints imaged.

We have added an overall schematic that will help readers orient themselves (Figure 1C) as well as a representative cutout example (Figure S6). We have also added a localization "call" for each gene in the table. All images are downloadable from our image repository and can be rotated by the user if this type of visualization is needed. We are working with Echinobase to link all localization data to gene accession number, and we hope this may provide additional visualization tools.

7. In discussion, we want to encourage the authors to point out the use of this high throughput method for single cell RNAseq data validation, when sometimes 100s of transcripts need to be analyzed. This method would make scRNAseq validation much less laborious.

Indeed. We now mention this.

REVIEWER 3.

1. Readers must view and personally interpret each of the FISH images shown, which each contain numerous embryos of mixed stages in random orientations. I think it would be a benefit to the reader to include an additional final figure that schematizes the overall findings, especially for novel gene expression profiles found in Figures 3-7. Embryo schematics for each timepoint with expression patterns painted on them would be ideal. These could potentially instead be added on a per-figure basis rather than collected in a final summary figure, and I'll leave it to the authors to choose how to approach this. This request applies to novel gene expression information; it does not seem necessary to me to also include spatial expression patterns that are already established.

We have added an overall schematic that will help readers orient themselves (Figure 1C) as well as a representative cutout example (Figure S6). We have also added a localization "call" for each gene in the table. All images are downloadable from our image repository and can be turned into painted schematics by the end user, if this type of visualization is needed. We are working with Echinobase to link all localization data to gene accession number, and we hope this may provide additional visualization tools.

2. In a related manner, Table S1 includes all the genes assessed, but lacks their results. I think it would be the best way to disclose the spatial outcomes of all the FISH experiments in the manuscript by listing the spatial patterns observed at each of the three assessed timepoints by adding an additional three columns to this table.

Done. We thank the reviewer for this suggestion.

3. I appreciate that it is lab jargon to call this approach "HCR", but this is vague and ambiguous. HCR stands for hybridization chain reaction (only), as we all know. There is an HCR approach for immunofluorescence labeling as well as one for FISH, both produced by Molecular Instruments. Just saying "HCR" is therefore ambiguous. It's fine for casual talk, but for a

publication, precise language is needed. Therefore, each instance of "HCR" should be extended to HCR-FISH. That includes the technique's name, which should be amended to HT-HCR-FISH.

Thanks for the suggestion. We decided not to use the suggested name, as it would be "High Throughput-Hybridization Chain Reaction-Fluorescent In Situ Hybridization" (which seemed redundant). In addition, there is nothing preventing anyone from adapting this protocol for antibody bound samples, and so we felt that HT-HCR appropriately alludes to how malleable the approach may be.

4. The methods section is missing some key information, including the species and source of animals used for the experiments shown and their husbandry, as well as the methods employed for manual FISH. Please explain the rationale both for the repeated cold shocks that the embryos are exposed to prior to fixation and for their deciliation prior to fixation. Each of these strikes me as unnecessary stresses to the embryos that could potentially influence their gene expression, particularly for stress-responsive genes. I see what was written about concentrating this way, but it is unclear to me why concentration by gravity (not literally by "ice", as written) was not performed following fixation to avoid the cold shock. I infer that icing the embryos blocks their swimming, causing them to settle, but this is not clearly explained and should be. The stages employed are described by their time post-fertilization; please also indicate what stage these timepoints correspond to.

We had included species information in the title of the paper "Automated, high-throughput in-situ hybridization of Lytechinus pictus embryos" and in the methods section, "Three independent batches were run for 24 hpf embryo samples, and three batches were run for mixed stage samples. For each batch of embryos, sperm from one male and eggs from two to four female L. pictus adults were spawned and mixed according to (Nesbit and Hamdoun, 2020)."

We added the other key pieces of information and made changes to clarify the embryo concentration section. The sample preparation process was adapted from the current protocol for performing the assay on sea urchin embryos from Molecular Instruments, which includes these steps. This protocol is available through this link: <https://files.molecularinstruments.com/MI-Protocol-RNAFISH-SeaUrchin-Rev10.pdf>. We added a key to the stages studied.

5. Regarding those genes that did not generate an interpretable signal, more consideration is needed, and this should be an aspect of the main results section rather than being shunted to supplemental figures and the discussion. Since ~ half the genes failed at this step, it seems that this method could be seen as half- successful by some readers; thus, an effort to understand why so many do not succeed is warranted.

Please see response to Reviewer 2 Comment 1. More discussion is now included.

6. Within the methods, please clarify the actual probe concentrations employed- are these uniform across all the probes? Was any testing done or were calculation performed to optimize probe concentrations? This is potentially relevant to those probes with negative results. In our hands, HCR-FISH is highly sensitive to probe dose, and this is only logical, since mRNAs are present across a very wide range of concentrations from gene to gene.

Thank you for pointing this out. We have added the final concentration in the methods. Regarding probe concentration we have provided additional data and discussion. The data didn't support the hypothesis adding more probe would resolve rare messages, presumably because probe is already in excess, and the target (mRNA) is the limiting factor (Figure S3). Without seeing the results the reviewer alludes to, we can't speculate on what is going on in their hands. However, it is conceivable that other assay conditions (eg imaging setup or other reagents) might be contributing factors.

7. Please include plots that compare the expression level at the stages of interest for the successful genes versus the unsuccessful ones to assess whether a higher dose could make a

difference. This point should be made in the results, the plot shown, and this point should also appear in the relevant part of the discussion.

It is well established that in situ hybridization is generally more effective for abundant transcripts than for rare ones. Our data do not support the idea that low-abundance targets can be resolved by increasing probe concentration (Figure S3).

In addition, transcript abundance is just one of several factors influencing detection success. While there is a trend toward better results with abundant targets, the assay is still capable of localizing low-abundance transcripts (e.g., SLC18), depending on their local concentration. Ultimately, target selection should be guided by the underlying biological question, and for some applications, probing for low-abundance gene products will be essential. A key advantage of this assay is that it streamlines evaluation of these targets, before investing in further work.

8. I would like to see a small set (~ 10) of those negative genes re-assayed with a higher concentration of the probes in a dose-response experiment that tests double, triple and quadruple the normal probe dose. I would select them over a range of gene expression levels rather than choosing a group with uniform levels of expression. Please explicitly clarify what these actual probe concentrations are and include this in a main figure.

Please see response to Reviewer 1 Comment 2 and above. We performed more experiments to address the reviewer's question. The results did not support the hypothesis that increasing probe dose will resolve rare genes (Figure S3). We only saw dose dependence for extremely abundant messages, that were already well visualized. This might reflect the higher molar ratio of target relative to probe for extremely abundant messages. We don't believe this warrants inclusion as a main figure, but we included the new data as a supplement (Figure S3).

9. The images from this work were collected, but no further analysis was performed. Please discuss this next component: how will quantification of these data be approached as the next necessary step?

In situ hybridization is primarily a qualitative technique, and we were unsure of specifically what type of quantification the reviewer was alluding to. Some studies show a "representative" image, and a count of how many embryos showed this pattern. This is especially important for perturbation studies where a range of phenotypes might have been produced. In this case we are imaging controls with common patterns of expression, and also directly showing multiple embryos for each target. In the future we expect that quantification of in situs may involve application of machine learning algorithms to image data. This study is a step towards this goal, as it facilitates rapid generation of model training data.

10. The citations in the first paragraph of the introduction do not represent the use of FISH in sea urchins. Only five papers are cited, all more than 10 years old, and only two, from more than 20 years ago, are offered as representative of "long history" of FISH in the field. The remaining three are focused very narrowly and are therefore not representative. The number of cited papers here should be increased to better represent the broad use of FISH in sea urchin developmental biology and should include recent as well as older papers.

We have added more references.

11. Figure 1's legend states that five sets of probes are used for each sample, but I think this should instead be four.

Thank you for catching this. We have fixed this issue. This was a mistake on our side, as we imaged with five wavelengths including the DAPI stain, which is of course not a probe set.

12. The first two sentences of the legend for Figures 4-7 are exactly the same. Please revise: The legend for Fig. 5-7 can say that the images were collected and annotated as in Fig. 4 (or

similar).

We have fixed this issue.

13. There is a lot of statement of results in the legends text that seems more appropriate for the main text or otherwise redundant to it. The legends can thus be more succinct.

We have previously been encouraged to make figure legends stand independently and prefer to keep them as is.

14. Supp Figure 2's legend states that tubulin alpha 1 is ubiquitously expressed, but the image shows clear enrichment in what seems to be the PMCs, the ciliary band, and the apical plate at the various timepoints shown.

We now mention this in the legend.

Bell, G. W., Yatskievych, T. A. and Antin, P. B. (2004). GEISHA, a whole-mount in situ hybridization gene expression screen in chicken embryos. *Dev. Dyn.* **229**, 677-687.

Nesbit, K. T. and Hamdoun, A. (2020). Embryo, larval, and juvenile staging of *Lytechinus pictus* from fertilization through sexual maturation. *Dev. Dyn.* **249**, 1334-1346.

Thisse, B., Heyer, V., Lux, A., Alunni, V., Degrave, A., Seiliez, I., Kirchner, J., Parkhill, J.-P. and Thisse, C. (2004). Spatial and temporal expression of the zebrafish genome by large-scale in situ hybridization screening. *Methods Cell Biol.* **77**, 505-519.

Tomancak, P., Beaton, A., Weiszmann, R., Kwan, E., Shu, S., Lewis, S. E., Richards, S., Ashburner, M., Hartenstein, V., Celniker, S. E., et al. (2002). Systematic determination of patterns of gene expression during *Drosophila* embryogenesis. *Genome Biol.* **3**, RESEARCH0088.

Tomancak, P., Berman, B. P., Beaton, A., Weiszmann, R., Kwan, E., Hartenstein, V., Celniker, S. E. and Rubin, G. M. (2007). Global analysis of patterns of gene expression during *Drosophila* embryogenesis. *Genome Biol.* **8**, R145.

Second decision letter

MS ID#: dev.204814R1

MS TITLE: Automated, high-throughput in-situ hybridization of sea urchin (*Lytechinus pictus*) embryos.

AUTHORS: Yoon Lee, Chloe Jenniches, Rachel Metry, Gloria Renaudin, Svenja Kling, Evan Tjeerdema, Elliot W. Jackson and Amro Hamdoun

Dear Dr Hamdoun,

I am happy to tell you that your manuscript has been accepted for publication in *Development*, pending our standard publication integrity checks.

Reviewer 1

Advance summary and potential significance to field

Lee et al. have addressed some of the concerns that I had with the previous version of the "Development Automated, high-throughput in-situ hybridization of sea urchin (*Lytechinus pictus*)

embryos" manuscript. In its current form, the quality of the data and downstream analyses seems overall very good and some additional controls have been added. The description of the methods is appropriate for a resource paper and some of the overreaching claims have been toned down when necessary. Yet, the responses provided by the authors have not completely lifted my concerns remain regarding the target audience of their methodology, which I still think is very niche for several reasons. First, most marine stations where sea urchins are used as model species are typically not equipped with the hardware required for liquid handling and automated imaging (Scripps is an exception among marine stations). Second, considerable work seems required to adapt this protocol to other hardware solutions (liquid handler or confocal brands) which are most likely to be found in other locations where users could be interested in applying a similar methodology. Finally, the protocol described in this manuscript is tailored to early developmental stages of sea urchins, but will require substantial modifications for use in either larger samples or different species. To sum this up, from my perspective of reviewer I think this manuscript is fit for publication regarding its technical aspects and rigorousness, but whether it is broadly applicable by the community and fits the scope of a journal like Development in terms of impact should be assessed by the editors.

Reviewer 2

Advance summary and potential significance to field

Authors have adequately addressed the comments.